# Spatial determinants of farmers' interest in European Union's pro-investment programs in Poland

Ewa Kiryluk-Dryjska[1], Barbara Więckowska[2], Arkadiusz Sadowski[1] *

**1** Faculty of Economics, Poznan University of Life Sciences, Poznan, Poland, **2** Department of Computer Science and Statistics, Poznan University of Medical Sciences, Poznan, Poland

* sadowski@up.poznan.pl

**Data Availability Statement:** All relevant data are within the manuscript and its Supporting Information files.

## Abstract

The purpose of the paper is to investigate spatial determinants of farmers' interest in pro-investment programs co-financed by the EU, by identifying and describing the territorial clusters of rural areas in Poland where the applications rates for these programs were above or below the national average. We tested for spatial autocorrelation using Moran's global spatial autocorrelation index, while the search for clusters was done using a local version of Moran's statistics. The results show significant regional variation in the farmers' interest in these programs in Poland. This interest was higher in regions with a greater level of agricultural development and better agrarian structure. In Poland, both of these factors are related not only to natural conditions, but also to strong historical context. We conclude that the pro-investment programs contribute to the deepening of development differences in Polish agriculture in the territorial dimension, which is not in line with the basic assumptions of cohesion policy.

## Introduction

Although, Poland participates in the Common Agricultural Policy (CAP) rural development policy since its accession to the EU in 2004, the financial perspective 2007–2013 was the first full-time UE programming period during which Poland could choose the programs from a full range of possible policy measures accessible for all EU countries. For the first three years of the membership, Poland implemented a Temporary Rural Development Instrument (TRDI). The program was designed to prepare new member states to meet UE standards in terms of agricultural production and to enhance structural changes in rural areas. The TRDI was targeted to support four accompanying measures (agri-environment, early retirement, afforestation, and LFA payments) and a number of small-scale measures specifically introduced for these countries, such as developing producer groups, support for semi-subsistence farms, technical assistance, and complements to direct payments [1]. Based on the general strategic guidelines for rural-development programs supported by the European Agricultural Fund for Rural Development, Polish government drew up the first 7-year Rural Development Program (PRDP 2007–2013) [2]. The program had been linked to the obligatory objectives imposed by

**Funding:** Initials of the authors who received each award: E. K-D Grant numbers awarded to each author: 2017/25/B/HS4/02513 The full name of each funder: National Science Centre, Poland URL of each funder website: http://www.ncn.gov.pl/?language=en The funders had no role in study design, data collection and analysis, decision to publish, or preparation of the manuscript.

**Competing interests:** The authors have declared that no competing interests exist.

the European Council Regulation (EC) 1698/2005: improving competitiveness of the agricultural and forestry sector, improving the environment and the countryside and improving the quality of life in rural areas and diversifying the rural economy. Additionally, the LEADER program was to be introduced in national or regional rural development programs of all member states. It is an initiative designed to build social capital through mobilization of rural population and to contribute to the creation of new jobs in rural areas, as well as to improve the management of local resources. Current Polish Rural Development Program for the period 2014–2020, is a continuation of the previous program. It was established by the Ministry of Agriculture and Rural Development, based on the EU Council Regulation No 1305/2013. It comprises a set of policy programs aiming at increasing the competitiveness of Polish agriculture, providing sustainable management of natural resources, and ensuring sustainable regional development of rural development.

While accessing the EU, Polish agriculture and rural areas were lagging behind in terms of productivity, overemployment, unfavorable farm structure and infrastructural underinvestment compared to old members countries [3]. With such a range of disadvantages, the structural programs were expected to meet many different rural development objectives. However, the need of farm modernization has been exposed the most. In years 2007–2013 over 42% of PRDP budget was allocated to improving competitiveness of the agricultural and forestry sector, whence 31% (2.2 billion Euros) for the two main pro-investment measures, namely 'modernization of agricultural holdings' and 'setting up of young farmers'. [1] outlines that the farm-centric bias of the rural development programs in new member states of the EU was enforced by granting the task of drawing up Rural Development plans to national Ministries of Agriculture. Indeed, despite strong regional differences in natural, economic, socio-cultural or technical conditions of Polish rural areas [4–6] planning of rural development programs in Poland remained centralized by national Ministry of Agriculture and Rural Development.

There is ongoing discussion in the literature on the impact of structural funds on regional differences in EU [7–13]. There is also no agreement concerning the impact of the CAP on the reduction of regional disparities. Some studies suggest its positive effects [14,15], others stress exactly the opposite [16,17]. Thus, there is an open question how the UE rural policy affects regional differentiation in Poland, where the development differences were initially strong.

There is an evidence in the literature showing that agricultural structures and local development conditions seem to have a large influence on expenditure in rural development programs [18–22]. However, as these effects differ across RDP measures, there is a need for further investigations on RDP distributive issues [20]. As emphasized by [18] 'specially disentangling RDP expenditures in single axes and measures may be particularly innovative to explain the real allocation of RDP funds and its effects throughout Europe'.

In this paper we analyze two pro-investment measures of PRDP 2007–2013, 'setting up of young farmers' and 'modernization of agricultural holdings', introduced in PRDP 2007–2013, under the first objective of the European Council Regulation (EC) 1698/2005. The purpose of the research is to investigate spatial determinants of farmers' interest in these programs by identifying and describing the territorial clusters of rural areas in Poland where the applications ratio for these programs was above or below the national average. The existence of such clusters might suggest that the policy toward modernization favors areas with specific structural features, which might enforce regional differentiation of rural areas in Poland.

The outline of the paper is as follows. First, we present the research method. Next, we shortly characterize analyzed pro-investment programs of the PRDP. Finally, we delimitate clusters of rural areas in Poland where farmers' interest in pro-investment programs is above or below the national average. Based on that we research the impact of local rural development determinants on the farmers' application and conclude with a discussion.

## Method

The research was conducted for Poland at the level of municipalities (NUTS 5 regions)—the smallest administrative units in Poland. Rural and rural-urban municipalities were taken into account (excluding urban ones). For each of them, the frequency coefficients of farmers' applications for the 'modernization of agricultural holdings' and 'setting up of young farmers' in the PROW 2007–2013 program were determined $r_i$. The pro-investment activities are also being implemented in the current EU financial perspective, however, due to the duration of the programme, no comprehensive application data is available. Therefore, the article focuses on the PROW 2007–2013 programme, for which complete summaries are available. The coefficients were calculated based on the number of applications submitted by farmers (separately for each of the analysed measure) compared to the total number of the farms for each municipality. It deliberately focuses on the number of submitted applications and not on the number of signed contracts, as it reflects better the interest of farmers in the scheme. Since the value of the coefficients may be influenced by the size of the population (the number of farms in a given municipality), some coefficients may be better estimated than others and this may distort the spatial distribution of frequencies. Coefficients for small population size may create artefacts reflecting a lack of sufficient data rather than actual frequency. To reduce this discrepancy, the calculated coefficients were applied with the empirical Bayesian smoothing proposed by [23], calculated using the following formula:

$$smooth(r_i)_{Bayes} = smooth(r_i) + c_i[r_i - smooth(r_i)]$$

where:

$smooth(r_i)_{Bayes}$ is the new Bayes smooth coefficient of frequency
$smooth(r_i)$ is the local weighted average, i.e. determined after joining direct neighbors
$c_i$ is the smoothing coefficient,
$r_i$ is the raw level of the coefficient of frequency.

This method is particularly useful when the coefficient instability results from a small size in some areas [24], while the large size areas are described by stable and unadjustable coefficients. The spatial analyses were based on the Queen adjacency matrix with elements $w_{ij}$, i.e. the assumption was made that the municipalities which share a border with a non-zero length are considered to be adjacent. The adjacency matrix was standardized so the rows sum to one in order to equalize the impact of municipalities with a large and small number of neighbors. The occurrence of spatial autocorrelation was checked by using Moran's global spatial autocorrelation index ($I$).

$$I = \frac{\sum_{i=1}^{n} \sum_{j=1}^{n} w_{ij}(smooth(r_i)_{Bayes} - \bar{r})(smooth(r_j)_{Bayes} - \bar{r})}{\left(\sum_{i=1}^{n} \sum_{j=1}^{n} w_{ij}\right)\left(\frac{\sum_{i=1}^{n} (smooth(r_i)_{Bayes} - \bar{r})^2}{n}\right)}$$

where:

$n$ –number of municipalities
$smooth(r_i)_{Bayes}$, $smooth(r_j)_{Bayes}$–Bayes smooth coefficients of frequency in $i$ and $j$ municipalities $\bar{r}$–global coefficient of frequency for Poland in the years covered by the analysis.

Low values of the coefficient ($I$) indicate the absence of autocorrelation, high (i.e. close to one) and statistically significant values of the Moran's global index reflect a positive autocorrelation, i.e. a strong tendency to build clusters.

The search for clusters involves merging the municipalities which are characterized by a higher/lower value of the analysed feature (the frequency of applying for a support as part of 'modernization of agricultural holdings' or 'setting up of young farmers') than the average

value for the whole analysed area. For this purpose, a local version of Moran's statistics, developed by [25] was used, with the following formula:

$$I_i = \frac{(smooth(r_i)_{Bayes} - \bar{r}) \sum_{i=1}^{n} w_{ij}(smooth(r_j)_{Bayes} - \bar{r})}{\frac{\sum_{i=1}^{n} (smooth(r_i)_{Bayes} - \bar{r})^2}{n-1}}$$

The local Moran's autocorrelation coefficient $I_i$ served as a basis for separating the statistically significant (at 0.05) cluster of above-average frequencies of applying for pro-investment support (High-High) from the cluster of below-average values (Low-Low) and outliers. The outliers are territories which statistically significantly differ from the adjacent municipalities in how they apply for pro-investment funds. If a municipality with a high level of the coefficient of applying for the support is adjacent to municipalities at low levels, it is designated as High-Low. In turn, if a municipality reporting a low level of the coefficient is surrounded by municipalities at high levels, it is labeled as Low-High.

To characterize municipalities where the frequency coefficients of farmers' applications for pro-investment measures was above or below the national average, synthetic features (factors) describing agriculture and rural areas in Poland, presented by [22] were used. The factors were determined using factor analysis based on 59 indexes describing agriculture and rural areas at the municipality level. Each of them consisted of several variables determined based on statistical data. They covered 70.4% of the overall data variance. The extracted factors were as follows: infrastructure, farm structures, organic agriculture, demography, animal production, entrepreneurship and agricultural land greening [22]. Table 1 presents factors with corresponding variables derived from factor analysis.

As factor scores for all municipalities sum up to zero, for individual municipalities, they reflect their relative position with regard to the level of the extracted features of agriculture and rural areas in Poland. In this study, we used factor scores to measure the level of the selected features of agriculture and rural areas in each municipality and in each cluster, as proposed by [26].

When characterizing the areas with high/low frequency coefficients of farmers' applications for the analysed activities, the High-Low outlier municipalities were combined with High-High clusters to form a group defined as High, and the Low-High outlier municipalities with Low-Low clusters to form a group defined as Low. The approach was carried out separately for each of the activities. Due to the existing obliqueness of the data obtained as a result of factor scores analysis, the characteristics of the areas belonging to the analyzed groups of municipalities were characterized using non-parametric methods. The Kruskal-Wallis test was used along with Dunn-Bonferroni's post-hoc method. For geospatial analyses, GeoDa v1.14 was used, other statistical analyses were performed in PQStat v1.6.8. The level of significance was set at 0.05.

## Pro-investment measures of PRDP 2007–2013

Among many activities aimed at the development of agriculture and rural areas, the European Union places a particular emphasis on co-financing of measures aimed at investments in the modernization of agricultural holdings. This is an issue of particular importance in the new EU member states, where delays in this respect have significantly reduced the competitiveness of the agricultural sector in a single market. Therefore, from the pre-accession SAPARD program to the current rural development programs, the measures aimed to finance investments in agricultural holdings occupy an important place in rural development programs. In the PRDP 2007–2013, there were two measures aimed to support the modernisation of farms:

**Table 1. Factors with corresponding variables derived from factor analysis.**

| Factor | Variables |
|---|---|
| *Farm structures* | Share of utilized agricultural area belonging to farms with 15 or more hectares (%) |
| | Share of farms with over 15 hectares of utilized |
| | Average farm size (hectares) |
| | Area of agricultural land per 1 tractor (hectares) |
| | Average sown area per farm (hectares) |
| | Share of industrial crops in total sown area (%) |
| *Infrastructure* | Primary schools (per 100 km2) |
| | Lower secondary schools (per 100 km2) |
| | Length of the sewage network (km per 100 km2) |
| | Length of the water supply network (km per 100 km2) |
| | Length of the gas supply network (km per 100 km2) |
| | Population connected to gas supply network in % of population |
| | Population per 1 km2 |
| *Organic agriculture* | Share of organic farms among total farms (%) |
| | Share of organic farms within total farm area(%) |
| *Demography* | Age dependency rate |
| | Number of children in comparison to elderly people |
| | Live births per 1000 women |
| | Real population increase (per 1000 people) |
| *Animal production* | Average herd size in a farm (in large heads) |
| | Size of poultry flocks in a farm (in units) |
| | Size of the cattle herd in a farm (in units) |
| *Entrepreneurship* | Bed places in tourist accommodation facilities per 100 people |
| | Entities of the national economy per 1000 people |
| | Entities of the national economy per 1000 people at working age |
| *Agricultural land greening* | Share of permanent grassland within agricultural land (%) |
| | Relation of forested land to agricultural land |
| | Consumption of NPK fertilizers per 1 hectare of agricultural land (in dt) |
| | Agricultural production space valorization index |

Source: Based on [22].

'modernization of agricultural holdings' and 'setting up of young farmers'. Their main goal was to support the modernisation of farms with adequate production potential. In 2007–2103, the support for "modernisation of agricultural holdings" could be applied for by a farmer (natural and legal person) who met the requirements related to professional qualifications and had a holding of an appropriate economic size, calculated in the form of the European Size Unit (minimum 4 ESU, i.e. 4800 euro Standard Gross Margin). The obtained funds could be used, among other things, for a construction or a modernisation of buildings, machines, or an establishment and a modernisation of orchards or perennial plantations. The aid took the form of a reimbursement of part of the investment expenditure incurred (generally 40% of the eligible costs). The maximum amount of the aid granted to a single beneficiary during the period of PROW implementation could not exceed PLN 300 thousand (i.e. around 77 thousand euro).

The beneficiaries of the 'Setting up of young farmers' in the PRDP 2007–2013 were natural persons up to the age of 40 who were starting their first agricultural activity and had the appropriate professional qualifications. Between 2007 and 2013, the bonus amounted to PLN 100 thousand and at least 70% of this amount had to be spent on investments, such as livestock,

**Table 2. Global Moran results for clusters of municipalities.**

| Measure | Moran global Index | p value |
|---|---|---|
| Setting up of young farmers | 0.62 | <0.001 |
| Modernization of agricultural holdings | 0.50 | <0.001 |

Source: Own calculations.

farmland and equipment purchases, building modernisation and the establishment of orchards. Taking over a farm with an agricultural area no smaller than the national average (approx. 10 ha) and no larger than 300 ha was an important condition for receiving the bonus.

Beside different characteristics of potential beneficiaries, and in consequence different eligibility criteria, the overall goal of both measures was to support farm modernisation and restructuring.

## Results

The value of global Moran's statistic were 0.62 and 0.5 (with $p<0.001$) respectively for the 'setting up of young farmers' and for the 'modernization of agricultural holdings' (Table 2). This indicates that the municipalities tend to create clusters with different frequencies of application for both of pro-investment measures. The result was statistically significant for each measure, nonetheless, 'setting up of young farmers' showed a stronger tendency to form clusters (higher value of the global Moran's statistic than for the 'modernization of agricultural holdings').

Table 3 shows the numbers of municipalities which form clusters at the above-average frequency coefficients of farmers' applications for pro-investment measures (High-High); municipalities which form clusters at below-average application (Low-Low), High-Low and Low-High outliers and out-of-cluster municipalities (at medium levels of the coefficient of frequency), calculated using the Moran's local statistic. Fig 1 the location of the designated clusters.

In the case of the 'setting up of young farmers', more than 37% of the analysed municipalities were in clusters, with 341 municipalities (15.9%) belonging to a cluster with the above-average application frequency (High-High), and 454 municipalities (21.2%) to one with a frequency below the national average (Low-Low). In total, less than 1.5% of the municipalities were outliers, with 20 municipalities which had below-average application frequencies for 'setting up of young farmers', but were adjacent to municipalities at higher levels (Low-High outlier). On the other hand, 10 municipalities showed much higher frequency coefficients than the surrounding municipalities (High-Low outlier).

In the case of the 'modernization of agricultural holdings', less than 32% of the analysed municipalities were in clusters, of which 251 (11.7%) belong to the cluster with the above-average application frequency (High-High) and 421 (19.7%) to the Low-Low cluster. Over 2% of the municipalities were outliers, of which 33 were Low-High and 13 were High-Low outliers. In both cases, the out of clusters group consisted of municipalities with an average value of the

**Table 3. Local Moran results for clusters of municipalities.**

| Measure | Cluster High-High | Cluster Low-Low | Outlier Low-High | Outlier High-Low | Out of clusters |
|---|---|---|---|---|---|
| Setting up of young farmers | 341 (15.9%) | 454 (21.2%) | 20 (0.9%) | 10 (0.5%) | 1317 (61.5%) |
| Modernization of agricultural holdings | 251 (11.7%) | 421 (19.7%) | 33 (1.5%) | 13 (0.6%) | 1424 (66.5%) |

Source: Own calculations.

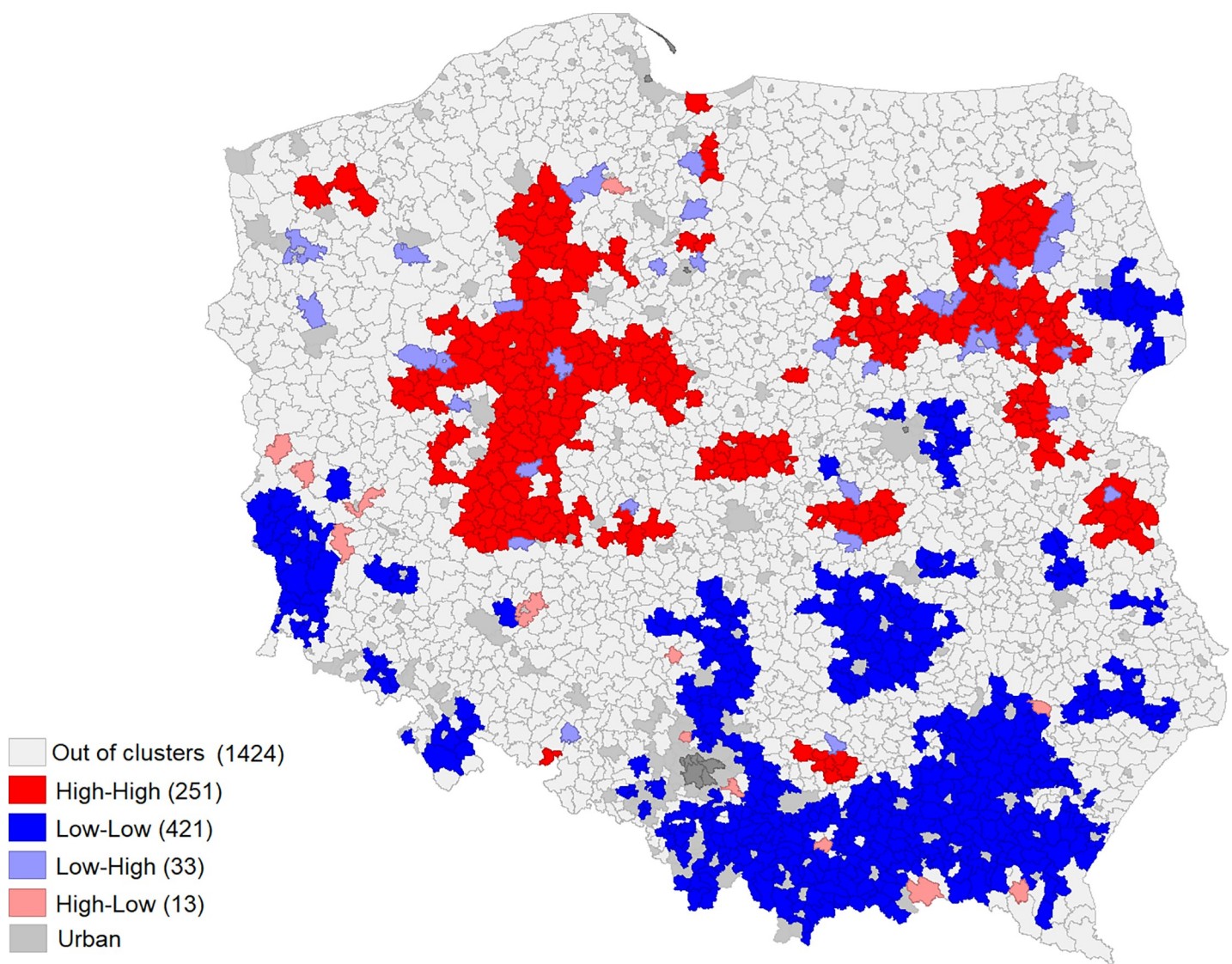

**Fig 1. Spatial location of clustered frequencies of applying for the measures 'setting up of young farmers' (left hand side) and 'modernization of agricultural holdings' (right hand side) based on the Moran's local statistic.**

frequency coefficient. Our results demonstrate that Low-Low and High-High dominate over outliers, which suggests that the territorial conditions affect the application ratios.

The results of local Moran's statistics confirm that the 'setting up of young farmers' shows a greater tendency to form clusters than 'modernization of agricultural holdings', both High-High and Low-Low clusters occur in this measure in greater numbers. There are also less municipalities in outliers than in the case of 'modernization of agricultural holdings'.

The spatial distribution of clusters of municipalities with above average and below average frequencies of applying for the 'setting up of young farmers' and 'modernization of agricultural holdings' shows similar pattern (Fig 1). Moreover, it strongly coincides with the territorial size structure of farms in Poland, expressed by the share of utilized agricultural area belonging to farms with 10 or more hectares (Fig 2).

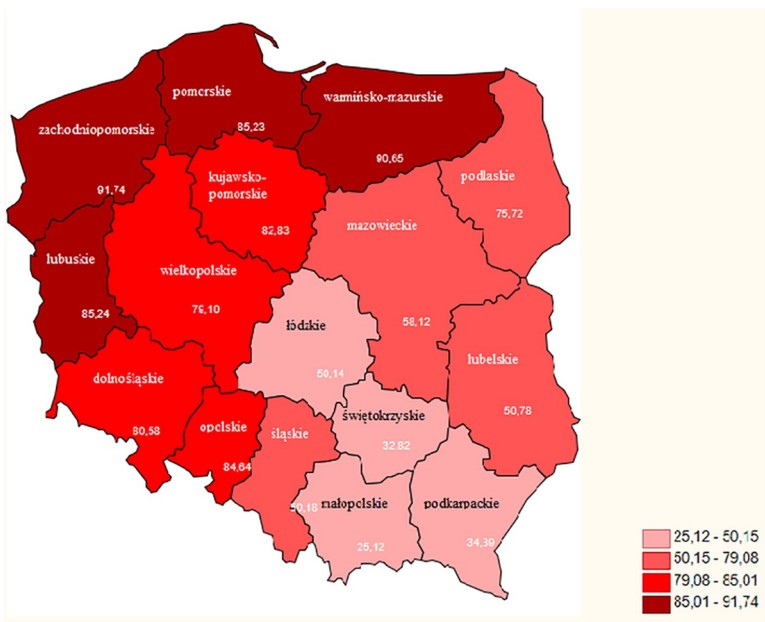

**Fig 2. Share of utilized agricultural area belonging to farms with 10 or more hectares in regions (Voivodeships) of Poland (%).** Source: Own elaboration.

In both cases, the concentration of municipalities belonging to the High-High cluster occurs in the west of the country, that is, in Wielkopolskie Voivodeship and Kujawsko-Pomorskie Voivodeship and in selected areas in the north-eastern Poland, the northern part of Mazowieckie Voivodeship and the western part of Podlaskie Voivodeship. A different situation occurs in the case of the Low-Low cluster, located mainly in the south-eastern part of Poland (in Małopolskie, Świętokrzyskie, and Podkarpackie Voivodeships). There are also smaller concentrations in the south-western part of Dolnośląskie Voivodeship and in the north-western part (Podlaskie Voivodeship). It is clearly visible that the interest in pro-investment programs is greater in areas, where the share of farms over 10 ha in the farm structure is relatively high. This applies primarily to the regions of the central-western part of Poland (Fig 2). The concentration of Low-Low municipalities is most intense in the south of the country, where the farm structure is most fragmented.

Table 4 presents the factor scores characteristic of municipality groups at the above-average application frequency for the 'setting up of young farmers' and 'modernization of agricultural holdings' (the HH cluster and HL outliers), and of municipalities at levels below the national average (the LL cluster and LH outliers). The results demonstrate that factor scores significantly differed between the municipality groups covered by the analysis. Figs 3 and 4 provide visualization of the factor scores for the municipality groups covered.

The Kruskall Wallis test result was statistically significant for all factor stores (p<0.0001). The results of the Dunn–Bonferroni post-hoc test show that all of the groups identified statistically significantly differ from each other in the three factors: *farm structures*, *infrastructure*, *and animal production*.

The highest values of the farm structures index were obtained by municipalities belonging to the High group (median factor scores are 0.11 in the case of 'setting up of young farmers' and 0.1 for 'modernization of agricultural holdings'). The lowest values of this index were recorded by the municipalities forming the Low group (median -0.4). Since the variables that are part of this factor (listed in the method) describe the quality of the agrarian structure, the

**Table 4. The values of factors scores characterizing the groups of municipalities with a higher (High) and lower (Low) than average value of the application indicator of the 'setting up of young farmers' and 'modernization of agricultural holdings' measures.**

| Factors | Type of cluster | 'setting up of young farmers' | | | | 'modernization of agricultural holdings' | | | |
|---|---|---|---|---|---|---|---|---|---|
| | | Median (Q1-Q2) | Post-Hoc Dunn-Bonferroni | | | Median (Q1-Q2) | Post-Hoc Dunn-Bonferroni | | |
| | | | High vs Low | High vs no cluster | Low vs co cluster | | High vs Low | High vs no cluster | Low vs co cluster |
| Farm structure | High | 0.11 (-0.31; 0.72) | <0.0001 | <0.0001 | <0.0001 | 0.10 (-0.36; 0.73) | <0.0001 | <0.0001 | <0.0001 |
| | Low | -0.4 (-0.80; -0.07) | | | | -0.38 (-0.74; -0.06) | | | |
| | no cluster | -0.26 (-0.71; 0.55) | | | | -0.25 (-0.71; 0.55) | | | |
| Infrastructure | High | -0.23 (-0.49; 0.02) | <0.0001 | <0.0001 | <0.0001 | -0.21 (-0.46; 0.12) | <0.0001 | <0.0001 | <0.0001 |
| | Low | 0.51 (-0.33; 1.59) | | | | 0.40 (-0.36; 1.56) | | | |
| | no cluster | -0.39 (-0.66; -0.04) | | | | -0.37 (-0.65; -0.02) | | | |
| Organic agriculture | High | -0.27 (-0.61; 0.07) | 1 | 0.000018 | <0.0001 | -0.33 (-0.67; -0.06) | 0.01357 | <0.0001 | 0.0014 |
| | Low | -0.29 (-0.62; 0.12) | | | | -0.24 (-0.61; 0.17) | | | |
| | no cluster | -0.18 (-0.45; 0.28) | | | | -0.18 (-0.47; 0.27) | | | |
| Demography | High | 0.13 (-0.51; 0.76) | 0.2731 | 0.000659 | <0.0001 | 0.06 (-0.51; 0.59) | 0.4401 | 0.6165 | 0.0008 |
| | Low | 0.17 (-0.34; 0.72) | | | | 0.14 (-0.45; 0.67) | | | |
| | no cluster | -0.11 (-0.72; 0.51) | | | | -0.05 (-0.67; 0.56) | | | |
| Animal production | High | 1.19 (0.6; 1.755) | <0.0001 | <0.0001 | <0.0001 | 1.05 (0.48; 1.76) | <0.0001 | <0.0001 | <0.0001 |
| | Low | -0.81 (-1.03; -0.48) | | | | -0.81 (-1.04; -0.48) | | | |
| | no cluster | -0.25 (-0.66; 0.34) | | | | -0.19 (-0.63; 0.48) | | | |
| Entrepreneurship | High | -0.33 (-0.79; 0.19) | 0.4368 | <0.0001 | <0.0001 | -0.24 (-0.73; 0.22) | 1 | 0.0007 | <0.0001 |
| | Low | -0.25 (-0.8; 0.36) | | | | -0.255 (-0.8; 0.33) | | | |
| | no cluster | -0.05 (-0.54; 0.54) | | | | -0.07 (-0.58; 0.50) | | | |
| Agricultural land greening | High | 0.13 (-0.44; 0.715) | <0.0001 | 0.0829 | <0.0001 | 0.28 (-0.26; 0.77) | <0.0001 | 1 | <0.0001 |
| | Low | -0.54 (-1.32; 0.30) | | | | -0.52 (-1.32; 0.30) | | | |
| | no cluster | 0.22 (-0.3; 0.8) | | | | 0.18 (-0.38; 0.77) | | | |

Kruskal-Wallis p-value<0.05

Source: Own calculations.

results indicate that the analysed measures were statistically more frequently applied by farmers from municipalities with better structural conditions of agriculture, characterized by greater than an average sown area and higher share of industrial crops.

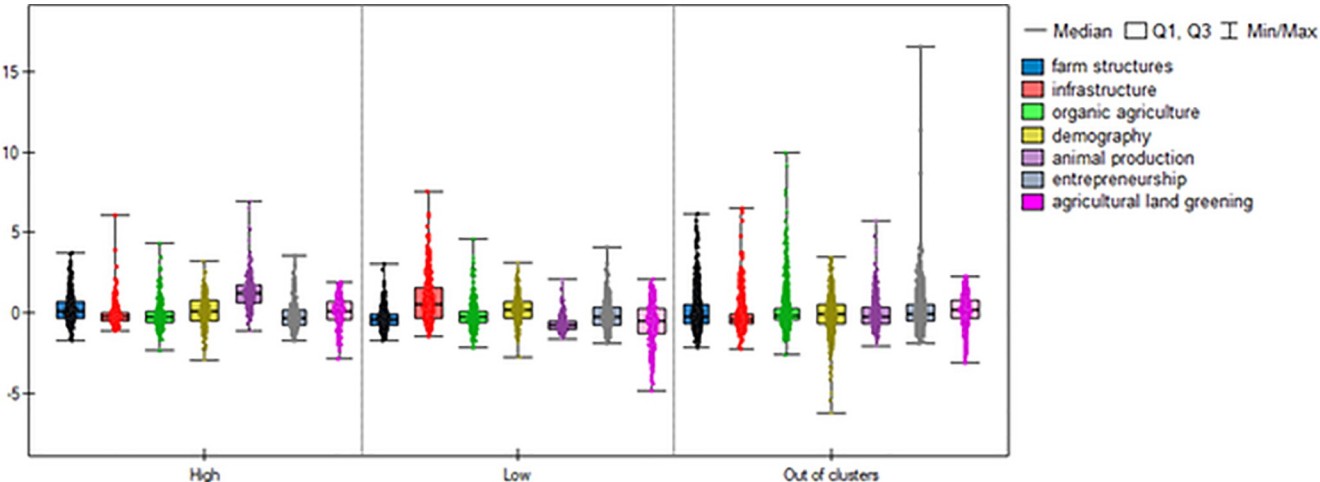

**Fig 3. Distribution characteristics of the factor scores for High, Low and Out-of-cluster groups for the measure 'setting up of young farmers'.** Source: Own elaboration.

Similar conclusions could be drawn from analysing the animal production factor. The value of the factor score for this factor in the municipalities of the high cluster was the highest (median 1.19 for 'setting up of young farmers' and 1.05 for 'modernization of agricultural holdings'), while the lowest rates were recorded in municipalities in the Low group (median in both groups of -0.81). This demonstrates that the regions with a high application rate for the analysed measures were characterized by a relatively good level of agricultural animal production compared to Poland. Other index that statistically significantly differentiated High vs. Low was also agricultural land greening. Its values were greater in the clusters with higher than average frequency of applications. The opposite relation can be noticed in the case of the infrastructure indicator. Its values were higher in the Low group for both analysed measures than in the remaining areas (median 0.51, upper quartile 1.59 for 'setting up of young farmers', median 0.40, upper quartile 1.56 for 'modernization of agricultural holdings'). Factors:

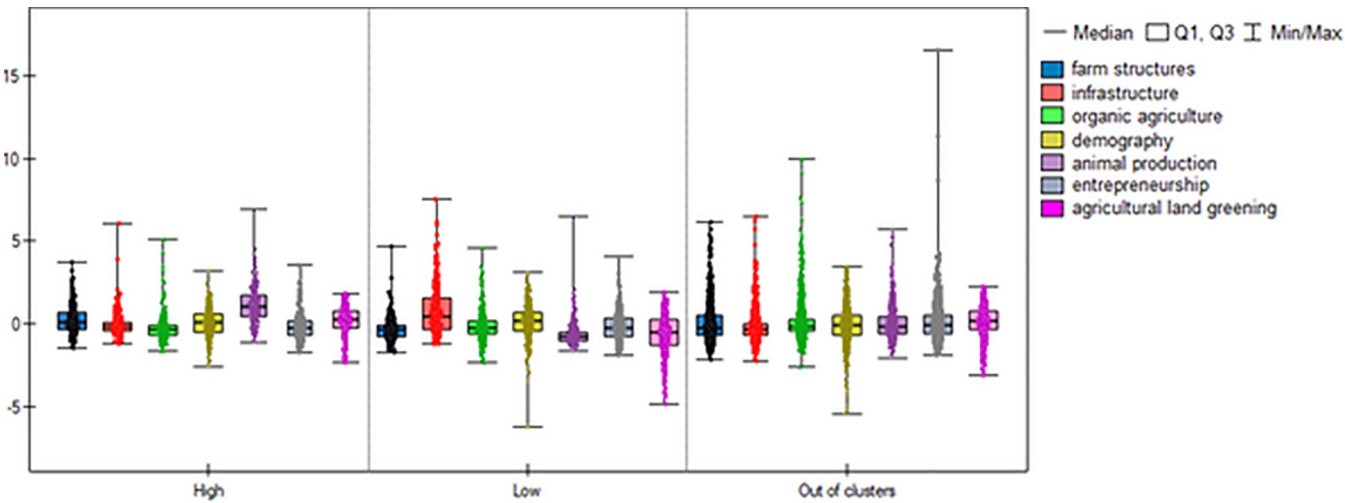

**Fig 4. Distribution characteristics of the factor scores for High, Low and Out-of-cluster groups for the measure 'modernization of agricultural holdings'.** Source: Own elaboration.

demography, organic farming and entrepreneurship and did not statistically significant differ-
entiate between clusters High and Low.

## Discussion

Our analysis indicated that the farm structure, animal production, and infrastructure were the
main factor differentiating selected clusters of municipalities (both for 'setting up of young
farmers' and 'modernization of agricultural holdings'). The above average high investment
activity occurs in municipalities with favourable agricultural structures and a predominance of
relatively large family farms. As one of the basic conditions for access to both 'setting up of
young farmers' and 'modernization of agricultural holdings' is to have a farm of a sufficient
size and an economic strength, this phenomenon is rather understandable. Undoubtedly, the
key factors that influenced the observed spatial structure of the support were the adopted
access criteria. The more entities meeting the boundary criteria of individual measures in a
given territorial unit, the greater interest of potential beneficiaries. Because of this, it is the
larger entities, predominantly located in municipalities belonging to the High-High cluster,
were more likely to take advantage of the funds. Similar results have been reported also in
other UE countries. [27] notes that investment activities in Latvia are carried out on larger
farms. [28,29] reached similar conclusions with regard to Danish agriculture. [30] emphasized
the importance of investment as an element necessary for the improvement of farm efficiency.

Such a distribution is justified, taking into account that the effective use of the investment
requires a sufficiently large farm in order not to disrupt the relationship between the land and
the capital, i.e. not to lead to an overinvestment. [31] showed that overinvestment, which does
occur in Polish agriculture, affects economically weaker farms to a greater extent, thus access
criteria for EU funds might mitigate this effect.

However, to have a wider overview of the possible geospatial effects of pro-investment
funds, it should be mentioned that the contemporary agrarian structure of Poland significantly
differs between regions and results from a number of socio-economic and political processes
that have taken place in the past [32]. In the 19th century, the territory of the present Poland,
divided between three partitioning countries (Russia, Germany and Austria) was subject to
different agricultural and economic policies. In the area of the German partition (mainly the
Wielkopolskie and Kujawsko-Pomorskie voivodeships), agriculture was supposed to be,
according to the assumptions of the authorities at that time, the food base for the industry
developing in the west of the country [33]. This sector has therefore developed rapidly. The sit-
uation was different in the south of present-day Poland, where the Austrian authorities did not
take action to counteract the division of farms, hence the structure is currently most frag-
mented. After the Second World War, there was a change of borders, including the takeover
by Poland of the eastern and northern areas of pre-war Germany (voivodeships: Warmińsko-
Mazurskie, Zachodniopomorskie, Pomorskie, Lubuskie, Dolnośląskie and Opolskie).

A significant diversification of the agrarian structure and consequently, the diversification
of the frequencies in applying for the modernisation funds is the contemporary effect of the
aforementioned processes. The highest concentration of the High-High clusters is in the cen-
tral-western part of Poland (Wielkopolskie and Kujawsko-Pomorskie voivodeships), with
large and family-run farms being dominant in the region. As noted by [34], the territorial
units located in what was once the German partition are generally more developed than the
rest of the country. To a smaller extent, this is also true of the northern part of the Mazowieckie
Voivodeship and the western part of the Podlaskie Voivodeship, even though in the 19th cen-
tury both of them were part of Russia. The situation in the areas taken over by Poland after the
World War II (western and northern areas) is different; contrary to the rest of the country,

where the communist authorities maintained the previous agrarian structure with the dominance of family farms, in the newly acquired areas they started to create large state farms. Following the political transformation of the 1990s, the state farms were later either commercialised and converted into companies, or divided between large family farms. These areas are dominated by legal entities, thus, despite the large farm size, their interest in investment-supporting instruments directed mainly at family farms is moderate. That is why in our analysis these areas are mostly in the out of clusters category for both analysed pro-investment measures. The Low-Low municipalities are mostly apparent in the south of the country, which was part of the Austrian partition, where, as already mentioned, the fragmentation of farms was not being counteracted. Moreover, this area consists mainly of mountainous and sub-mountainous areas that generally do not facilitate large farm concentration.

It is interesting that the clusters with below average frequency of application for the pro-investment measures were characterized by a higher level of infrastructure index. As this index. indicates the level of a development of technical (density of gas supply network and the sewage network) and social (density of schools, libraries, pharmacies etc.) infrastructure in rural areas, its high values occur in case of municipalities with a large population density. In Poland, those are mainly southern areas, while in agricultural regions with lower population density the infrastructure is less developed. Thus, the level of agrarian structures in Poland does not positively correlate with the infrastructure index.

Different interpretation applies to the link between the higher frequency of the applications for pro-investments funds and the natural factors, presented as the quality of agricultural land greening. From the microeconomic point of view, development farms try to purchase land of the highest possible quality as this guarantees to achieve the ground rent. According to [35], large and economically strong farms (regardless of a location) are generally characterized by better soil quality. It should be noted that, where larger farms predominate, agricultural culture and agricultural productivity are usually higher. Moreover, large family farms are particularly interested in the quality of agricultural production space (i.e. de facto a sustainable development), hence their users are interested in the protection of landscape elements such as meadows or forests, which ultimately impacts the production parameters of the soil on the farm itself (by, for example, regulating water relations). Moreover. a higher productivity allows excluding weaker land and limits afforestation.

The second pillar of the CAP, as an element of a EU structural policy, is expected to contribute to the overall objective of cohesion policy, which is to reduce the development differences between the EU areas. Thus, the stronger investment support in the municipalities, which already present a high level of agricultural development leads to deepening the development differences, and appears to contradict the core philosophy of the cohesion policy. These effects could be mitigated through other PRDP activities. However, the results of [22] and [26] show that such activities as farm diversification and micro-enterprises, which are aimed at changing the employment structure in rural areas by providing access to activity forms other than agriculture, are also of higher interest to potential beneficiaries in regions with higher levels of agricultural development. Similar results were presented by [36] in relation to agritourism. They demonstrated that obtaining funding for this type of activity is more frequent in areas where larger family farms predominate, rather than in areas featuring special natural or cultural values. This is also confirmed by the studies by [37], who note that positive development processes in agriculture and the countryside are mainly visible in areas already characterized by more favourable parameters. In the context of widening regional development differences, the presented effects of pro-investment policies are worrisome and tend to confirm the concerns previously notified by [16,17].

Our results also identified outlier municipalities. Low-High outliers were municipalities which, despite favorable development conditions and territorial proximity of municipalities of the High cluster, recorded below-average levels of activity in applying for the measure concerned. Conversely, High-Low outliers were municipalities which, despite a difficult structural situation and proximity of the Low cluster, demonstrated outstanding performance in applying for the measure. The existence of these outliers demonstrates that despite the determinants for applying for pro-investment programs specified in this paper (such as the agricultural and rural development level), the applicants may be driven by a series of other unrelated factors. This opens up a field for further detailed analysis of the specific factors stimulating the applications for pro-investment funds in the outlier municipalities. However, such research requires the methodology to be expanded to include analyses in sociology and political science. As demonstrated by [38] 'agricultural programs can result in positive horizontal spillover effects as they encourage participants to adopt a certain technology and hope that this will induce further adoption within the community or in neighbouring communities'. [39] suggest that factors such as social capital endowment and leadership strength, together with institutional and governance quality might influence the contrasting socio-economic performances of neighboring territories.

The presented method of merging the areas which are characterized by a high/low application ratio for selected programs might be used for other structural programs, not only in Poland but also in other EU countries.

Interestingly, the 'setting up of young farmers' shows a greater tendency to form clusters than 'modernization of agricultural holdings' (higher values of Moran's statistics and fewer outliers). The stronger territorial coherence in applications for 'setting up of young farmers' than for 'modernization of agricultural holdings' might suggest that young farmers are more susceptible for positive spillover effects. However, further studies would be needed to confirm this assumption.

## Conclusions

In summary, defining clusters with different frequencies of applying for pro-investment measures under PRDP showed significant regional differentiation in the farmers' interest in these programs. The interest was higher in regions with a greater level of agricultural potential and a better agrarian structure, which in the case of Poland is related not only to natural conditions but, above all, to historical factors. The results suggest that pro-investment programs contribute to the deepening of development differences in Polish agriculture in the territorial dimension.

The results showed that the spatial differentiation of applying for pro-investment funds was induced by historical factors, thus the direct conclusions of the paper refer specifically to Poland. Nevertheless, it has been also indicated that the differentiation of farmers' interest in the EU support was strongly determined by the agrarian structure. Thus, the obtained results have a more universal dimension. The presence of relatively large and economically strong family farms, capable of absorbing the funds and using them effectively, affects the application ratio. This finding, independent of the specificity of Poland, may constitute a premise for constructing the assumptions of agricultural and structural policy.

The studies of [18–21,40], demonstrate that also in other EU countries, in remote and lower income locations where rural development programs are most needed, the programs' performance is often relatively modest. However, it should be emphasised that we focused only on one specific group of programs which are designed for farm modernization. The other measures of the CAP not analysed in this study, and the cohesion policy might potentially

balance off the effects of pro-investment measures. Thus, to ensure the social and economic development of rural areas in regions of less favorable agrarian structures where agriculture is currently unable to compete, it would be critical to enhance conditions for alternative types of economic activities. This problem is particularly important in those areas where the structural deficits of the agricultural sector coexist with the failure of other sectors of the economy, contributing both to the deepening of the deprivation in rural areas and to the divergence of the level of socio-economic development. Although, the pro-investment support may contribute to the improvement of the income situation and living standard of farmers, the results show that this phenomenon occurs in areas where agriculture already has a strong position. This calls for the EU intervention which is well-tailored to the local needs. Otherwise the regional disparities between rural areas of Poland will only continue to deepen in direct contradiction of the general principles of the cohesion policy.

## Supporting information

**S1 Table.**
(PDF)

## Author Contributions

**Conceptualization:** Ewa Kiryluk-Dryjska.

**Data curation:** Ewa Kiryluk-Dryjska.

**Formal analysis:** Ewa Kiryluk-Dryjska, Barbara Więckowska.

**Funding acquisition:** Ewa Kiryluk-Dryjska.

**Investigation:** Ewa Kiryluk-Dryjska.

**Methodology:** Ewa Kiryluk-Dryjska, Barbara Więckowska.

**Writing – original draft:** Arkadiusz Sadowski.

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
