## [Decision Letter · Decision Letter 0]

9 Dec 2020

PONE-D-20-34284

Spatial determinants of farmers’ interest in European Union’s pro-investment programs in Poland

PLOS ONE

Dear Dr. Sadowski,

Thank you for submitting your manuscript to PLOS ONE. After careful consideration, we feel that it has merit but does not fully meet PLOS ONE’s publication criteria as it currently stands. Therefore, we invite you to submit a revised version of the manuscript that addresses the points raised during the review process.

We look forward to receiving your revised manuscript.

Kind regards,

Bing Xue, Ph.D.

Academic Editor

PLOS ONE

Journal Requirements:

2. We note that Figure 1 and 2 in your submission contain map images which may be copyrighted. All PLOS content is published under the Creative Commons Attribution License (CC BY 4.0), which means that the manuscript, images, and Supporting Information files will be freely available online, and any third party is permitted to access, download, copy, distribute, and use these materials in any way, even commercially, with proper attribution. For these reasons, we cannot publish previously copyrighted maps or satellite images created using proprietary data, such as Google software (Google Maps, Street View, and Earth). For more information, see our copyright guidelines: http://journals.plos.org/plosone/s/licenses-and-copyright.

(1) You may seek permission from the original copyright holder of Figures 1 and 2 to publish the content specifically under the CC BY 4.0 license. 

Reviewers' comments:

Reviewer's Responses to Questions

**Comments to the Author**

1. Is the manuscript technically sound, and do the data support the conclusions?

Reviewer #1: Yes

2. Has the statistical analysis been performed appropriately and rigorously? 

Reviewer #1: Yes

3. Have the authors made all data underlying the findings in their manuscript fully available?

Reviewer #1: Yes

4. Is the manuscript presented in an intelligible fashion and written in standard English?

Reviewer #1: Yes

5. Review Comments to the Author

Reviewer #1: The article presents an interesting study of farmers' interest in EU pro-investment programs.

The article has a local dimension, can these results be extrapolated to the international dimension? Could you describe it? The local perspective is interesting, but is it possible for other countries in the EU or in the world to adapt it? How to use this knowledge?

Improving the economic situation of farmers also means increasing the standard of living and quality of life. So can you answer: How do investment programs affect farmers' standard of living? The social part of sustainable development is worth pointing out. It will allow us to take a broad look at the problem under investigation.

When you look at Figure 1, you can see that two extreme situation - low-low and high-high. This groups dominate. What does that mean? Could you explain it?

The discussion is extremely interesting, but is it not worthwhile to make recommendations for state policy, and perhaps the EU as well?

It difficult to find the relationship between Figure 1 and 2, could you explain clearly?

I consider this article to be important. It is worth inserting it in the journal with minor revision.

6. PLOS authors have the option to publish the peer review history of their article (what does this mean?). If published, this will include your full peer review and any attached files.

Reviewer #1: No

---

## [Author Response · Author response to Decision Letter 0]

15 Jan 2021

We thank the reviewer for her/his constructive critiques. We feel the manuscript is greatly improved by incorporating their suggestions. Please find our detailed responses below:

1. The article has a local dimension, can these results be extrapolated to the international dimension? Could you describe it? The local perspective is interesting, but is it possible for other countries in the EU or in the world to adapt it? How to use this knowledge?

Thank you for bringing up this point. Although this research concerns Poland, its results contribute to a broader discussion on the impact of local factors on the effects of EU's structural policies. The studies quoted in the paper (lines 452-454) demonstrate that also in other EU countries, in remote and lower income locations where rural development programs are most needed, the programs’ performance is often relatively modest. Thus, it can be concluded that negative selection demonstrated in the paper, is taking place in case of rural development programs not only in Poland but in many EU countries. Moreover, the paper aims to contribute to the discussion on the impact of structural funds on regional differences in EU (lines 72-76 and also 444-454). Our findings suggest that pro-investment programs contribute to the deepening of development differences in Polish agriculture in the territorial dimension, which may be an important hint in for UE policy designing. It has been also indicated that the differentiation of farmers’ interest in the EU support was strongly determined by the agrarian structure. This finding, independent of the specificity of Poland, may constitute a premise for constructing the assumptions of agricultural and structural policy. The discussion of this point has now been added to the conclusions. 

Finally, in the paper we propose a method of merging the areas which are characterized by a high/low application ratio for selected programs, which may be used for analyzing of other structural programs in only in Poland but in other EU countries. Thank you for this remark, it has been now added to the paper (lines 427-429).

2. Improving the economic situation of farmers also means increasing the standard of living and quality of life. So can you answer: How do investment programs affect farmers' standard of living? The social part of sustainable development is worth pointing out. It will allow us to take a broad look at the problem under investigation.

Measures ‘modernization of agricultural holdings’ and ‘setting up of young farmers’ were implemented under the 1rd goal of the 2007–2013 RDP to ‘improve the competitiveness of the agricultural and forestry sector.’ Thus, in Polish conditions their main goal was to support the modernization of farms with adequate production potential. In 2007–2013 RDP, these programs were supplemented by the measures of objective 3, which directly aimed at increasing the standard of living and quality of life. However, as farm modernization results in overall better economic situation of farms’ holders, it also positively impacts the standard of living. Therefore, the conclusions of the research were extended to consider the social effects of the implementation of the discussed instruments, indicating that they contribute to the development of agribusiness and the standard of living of farmers mainly in areas where this sector is already strong (lines 463-465). Therefore, the need to apply other, pro-development forms of support in areas with a dysfunction of both agriculture and other sectors of the economy was indicated.

3. When you look at Figure 1, you can see that two extreme situation - low-low and high-high. This groups dominate. What does that mean? Could you explain it? 

In the paper we used the local Moran’s autocorrelation coefficient as a basis for separating the statistically significant cluster of above-average frequencies of applying for pro-investment support (High-High) from the cluster of below-average values (Low-Low) and outliers. These are the most important area of our investigation. However, we also delimitated the outliers (with a high level of the coefficient of applying for the support but adjacent to municipalities at low levels- High-Low, and with a low level of the coefficient surrounded by municipalities at high levels- Low-High). Our results demonstrate indeed that Low-Low and High-High dominate over outliers. This confirms that the territorial conditions affect the application ratios (lines 238-239). It is especially evident in Poland where agriculture structures differ regionally. However, the existence of the outliers (even if their number is limited) demonstrates that despite the determinants for applying for pro-investment programs specified in this paper (such as the agricultural and rural development level), the applicants may be driven by a series of other unrelated factors (explained in lines 415-426).

4. The discussion is extremely interesting, but is it not worthwhile to make recommendations for state policy, and perhaps the EU as well? 

In the paper we conclude that in order to ensure the development of rural areas in regions of less favourable agrarian structures where agriculture is currently unable to compete, it would be critical to enhance conditions for alternative types of economic activities. This calls for the EU intervention which is well-tailored to the local needs. The discussion on this point is in the last paragraph of the paper. 

5. It difficult to find the relationship between Figure 1 and 2, could you explain clearly? 

Figure 1 shows the spatial differentiation of the use of pro-investment measures in Poland, while Figure 2 shows selected aspects of Polish agrarian structure. One of the most important aspects of the research was to show the similarities between the location of high-high municipalities (Figure 1) and the share of farms with an area of more than 10 ha (Figure 2). The comparison of both figures was made in the Results section of the paper (lines 270-274). The dominant importance of structural factors in applying for EU funds has been also discussed in conclusions. 

Sincerely,

Ewa Kiryluk-Dryjska, 

Arkadiusz Sadowski (corresponding author)

Barbara Więckowska

---

## [Decision Letter · Decision Letter 1]

19 Feb 2021

Spatial determinants of farmers’ interest in European Union’s pro-investment programs in Poland

PONE-D-20-34284R1

Dear Dr. Sadowski,

We’re pleased to inform you that your manuscript has been judged scientifically suitable for publication and will be formally accepted for publication once it meets all outstanding technical requirements.

Kind regards,

Bing Xue, Ph.D.

Academic Editor

PLOS ONE

Additional Editor Comments (optional):

Reviewers' comments:

Reviewer's Responses to Questions

**Comments to the Author**

1. If the authors have adequately addressed your comments raised in a previous round of review and you feel that this manuscript is now acceptable for publication, you may indicate that here to bypass the “Comments to the Author” section, enter your conflict of interest statement in the “Confidential to Editor” section, and submit your "Accept" recommendation.

Reviewer #1: All comments have been addressed

2. Is the manuscript technically sound, and do the data support the conclusions?

Reviewer #1: Yes

3. Has the statistical analysis been performed appropriately and rigorously? 

Reviewer #1: Yes

4. Have the authors made all data underlying the findings in their manuscript fully available?

Reviewer #1: Yes

5. Is the manuscript presented in an intelligible fashion and written in standard English?

Reviewer #1: Yes

6. Review Comments to the Author

Reviewer #1: The article meets the requirements of the publisher. It is interesting and makes a valuable contribution to science. The authors have corrected the issues that bothered me and explained them sufficiently. There may have been little literature on the social side, but the all article is correct.

7. PLOS authors have the option to publish the peer review history of their article (what does this mean?). If published, this will include your full peer review and any attached files.

Reviewer #1: **Yes: **Sławomir Kalinowski

---

## [Editor Report · Acceptance letter]

23 Feb 2021

PONE-D-20-34284R1 

Spatial determinants of farmers’ interest in European Union’s pro-investment programs in Poland 

Dear Dr. Sadowski:

I'm pleased to inform you that your manuscript has been deemed suitable for publication in PLOS ONE. Congratulations! Your manuscript is now with our production department. 

Kind regards, 

on behalf of

Professor Bing Xue 

Academic Editor

PLOS ONE